# Translational coupling via termination-reinitiation in archaea and bacteria

Madeleine Huber[1], Guilhem Faure [2,3], Sebastian Laass[1], Esther Kolbe[1], Kristina Seitz[1], Christina Wehrheim[1], Yuri I. Wolf [2], Eugene V. Koonin [2] & Jörg Soppa [1]

The genomes of many prokaryotes contain substantial fractions of gene pairs with overlapping stop and start codons (**ATG**A or TG**ATG**). A potential benefit of overlapping gene pairs is translational coupling. In 720 genomes of archaea and bacteria representing all major phyla, we identify substantial, albeit highly variable, fractions of co-directed overlapping gene pairs. Various patterns are observed for the utilization of the SD motif for de novo initiation at upstream genes versus reinitiation at overlapping gene pairs. We experimentally test the predicted coupling in 9 gene pairs from the archaeon *Haloferax volcanii* and 5 gene pairs from the bacterium *Escherichia coli*. In 13 of 14 cases, translation of both genes is strictly coupled. Mutational analysis of SD motifs located upstream of the downstream genes indicate that the contribution of the SD to translational coupling widely varies from gene to gene. The nearly universal, abundant occurrence of overlapping gene pairs suggests that tight translational coupling is widespread in archaea and bacteria.

[1] Goethe University, Institute for Molecular Biosciences, Max-von-Laue-Str. 9, D-60438 Frankfurt, Germany. [2] National Center for Biotechnology Information, National Library of Medicine, National Institutes of Health, Bethesda, MD 20894, USA. [3] Broad Institute of MIT and Harvard, Cambridge, MA 02142, USA. Correspondence and requests for materials should be addressed to J.S. (email: soppa@bio.uni-frankfurt.de)

In most archaea and bacteria, a substantial fraction of genes is organized in operons, which are transcribed into polycistronic transcripts from a common promoter. Several potential selective factors for the evolution of operons have been considered[1–5].

It has been shown that translation is initiated independently on the genes on polycistronic transcripts, and the efficiencies depend on the corresponding individual translation initiation regions (TIRs) (Fig. 1a). Independent translation initiation at downstream genes in operons has been experimentally demonstrated for several examples and has been proposed to be widespread[6]. In addition, in vivo structural analysis of *E. coli* transcripts has shown that the extent of RNA structure can differ for different genes of the same transcript, indicating that genes in polycistronic mRNAs can have different translational efficiencies[7].

However, an advantage of polycistronic mRNAs is the possibility of translational coupling of two or more genes, whereby translation of a downstream gene depends on the translation of the upstream gene. About 30 years ago, it was discovered that translation of the downstream gene of the tryptophan biosynthesis operon, *trpA*, is coupled to the translation of the upstream gene *trpB*[8]. Subsequently, translational coupling has been described for various additional gene pairs in *E. coli* operons, for example, *atpHA*[9], *rplJL*[10], *motAB*[11], and *infC-rpmL*[12]. The mechanistic basis for translational coupling often involves base-pairing or pseudoknot formation of the TIR of the downstream gene with part of the open reading frame (ORF) of the upstream gene[9,13,14]. A long-range secondary structure is formed that prevents binding of the 30S ribosomal subunit to the TIR and thereby inhibits translation initiation. Translation of the upstream gene destroys this secondary structure and thereby acts as a switch to induce translation of the downstream gene (Fig. 1b).

Because translation is initiated by newly recruited 30S subunits, we hereafter refer to this process as "upstream translation-dependent de novo initiation" (UTNI). The UTNI mechanism strictly couples translation of the downstream gene to that of the upstream gene. However, the two genes have independent TIRs, and the relative strengths of these signals determine the levels of the two proteins. The level of the downstream protein can be either much higher or much lower than that of the upstream protein. For example, coupled translation of the *rplJL* gene pair results in a fivefold excess of RplL compared to RplJ[12], while, in contrast, coupled translation of *atpEF* results in an about tenfold lower level of AtpF than AtpE[15].

A mechanistically distinct mode of translational coupling is known as "termination-reinitiation" (TeRe). In this case, the same ribosome (or at least the same small subunit) that terminates at an upstream gene initiates translation at a nearby or overlapping downstream gene (Fig. 1c). The TeRe mechanism has mostly been studied in eukaryotic RNA viruses[16,17]. After translation of the upstream gene, the ribosome terminates and the large subunit dissociates from the mRNA, but the small subunit remains associated with the mRNA. Termination upstream ribosomal binding sites (TURBs) of ~70 nt are essential for retaining the small subunit at the mRNA, near the start codon of the downstream gene. The retained small subunit then recruits a new 60S subunit and translation is reinitiated.

In bacteria, it is unclear whether the 30S subunit or the 70S ribosome is involved, but TeRe has been proposed to operate. The relevant experimental evidence includes the demonstration that (1) ribosomes that terminate translation at an artificial ORF-internal stop codon can reinitiate at a nearby ORF-internal methionine codon[18], and (2) insertion of an AUG codon close to a stop codon redirects initiation from the native start codon to the closest AUG, which was interpreted as post-termination scanning by ribosomes[19]. The distances between stop codons and downstream start codons that appear to comprise the "reinitiation window" differ between reports. Although, to the best of our knowledge, there have been no systematic studies on the reinitiation window, it can be expected that the efficiency of reinitiation negatively correlates with the distance between the stop and restart codons, given that reinitiation requires scanning of the mRNA by the 30S subunit (or the 70S ribosome), and the probability of dissociation increases with the distance.

The smallest possible distance between the stop and start codons is an overlap, whereby the stop codon and the start codon share the same nucleotides. Examples are *UGA*UG (stop codon in italics, start codon in bold), which represents a gene overlap of one nucleotide, and **AUG**A, which is a four nucleotide overlap.

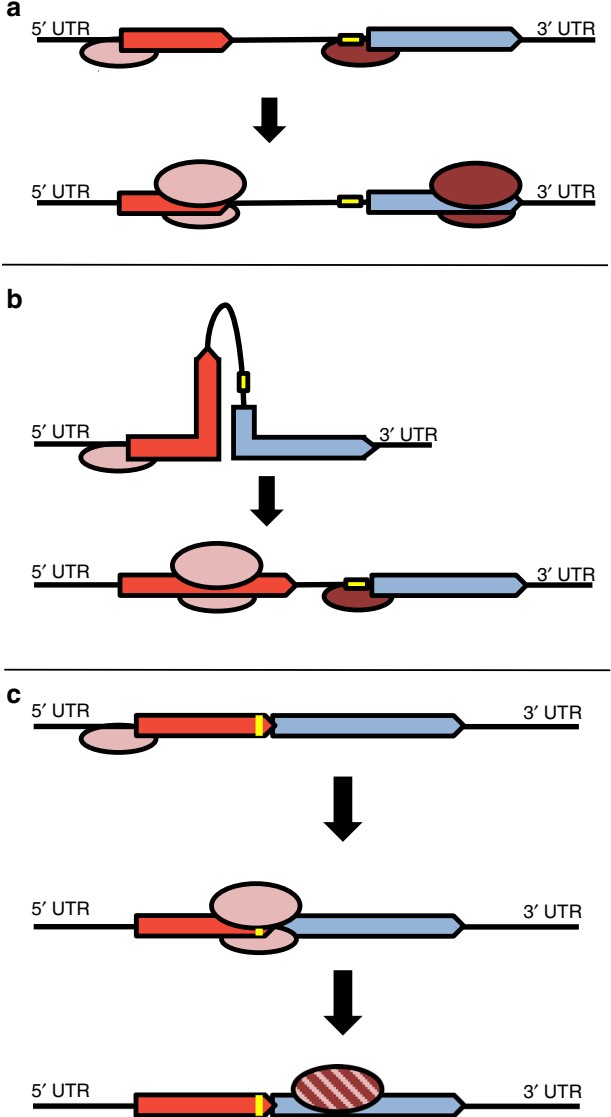

**Fig. 1** Three mechanisms of translation initiation at polycistronic mRNAs. **a** No translational coupling. Genes with intergenic regions, independent translation initiation at all genes. **b** Translational coupling (UTNI mechanism). Genes with intergenic regions. Translation of the downstream gene is inhibited due to long-range interactions. Translation of the upstream gene destroys these structures and enables do novo initiation at the downstream gene. **c** Translational coupling (TeRe mechanism). Genes with overlaps of stop and start codons. The same ribosome that terminates translation of the upstream gene (or the same 30S subunit) restarts translation at the downstream gene

Overlapping gene pairs are not rare in prokaryotes, and the PairWise database enables the analysis of individual genomes for the presence and fractions of overlapping gene pairs in all possible orientations[20]. Co-directional overlaps are more common than convergent or divergent ones, and the 4-nt overlap is the most common[20]. Analysis of 198 genomes has shown a higher degree of conservation for overlapping genes compared to non-overlapping genes[21]. Analysis of several parameters of overlapping genes suggests that selection for reduction of genome size is unlikely to drive the evolution of gene overlaps. Instead, it has been proposed that gene overlaps are involved in the regulation of gene expression[20].

The Shine-Dalgarno (SD) motif in a mRNA can base-pair with the 3′-end of the 16S rRNA and is found in the 5′UTRs of numerous genes in diverse bacteria, in particular, the majority of the *E. coli* genes[20]. The SD sequence has been shown to play an important role in the translation initiation of many bacterial genes[22–24]. Because SD is important at non-overlapping gene pairs in *E. coli*[25], it appears likely that this motif also contributes to the translation of overlapping gene pairs.

Here, we present a bioinformatics analysis of 720 archaeal and bacterial genomes. Most of these genomes contain substantial fractions of overlapping gene pairs in which the downstream genes are typically preceded by a (strong) SD motif. Experimental study of 14 overlapping gene pairs from the model archaeon *H. volcanii* and the model bacterium *E. coli* demonstrates tight translational coupling, but also reveals a highly variable role of the SD in this phenomenon.

## Results

**Distribution of overlapping gene pairs and intragenic SD motifs.** We first computationally analyzed the content and organization of overlapping gene pairs in 720 bacterial and archaeal genomes (see Methods). The species were selected, on the one hand, to represent all major groups of prokaryotes, and, on the other hand, to avoid oversampling of heavily studied groups. For each genome, we identified the co-directional overlapping gene pairs (with any length of the overlap) with potential for translational coupling (Supplementary Data 1). Figure 2 shows the mean fractions of co-directional overlapping genes pairs in 24 groups of prokaryotes. In most of these groups, the fraction of such gene pairs is close to 15%. *Cyanobacteria*, have <10%, four groups have a fraction >20%, and the highest fraction of overlaps, ~30%, was observed in *Aquificae*. Thus, ~30% of the genes in most prokaryotes and ~60% in *Aquificae* are organized in co-directional overlapping gene pairs.

For comparison, we estimated the fractions of "leading genes", i.e. those that are separated by at least 200 nt from the ends of the

corresponding upstream genes, so that translational coupling can be ruled out, and de novo initiation is required. The fractions of leading genes are typically ~40%, ranging from 25% in *Aquificae* to more than 50% in *Cyanobacteria* (Supplementary Fig. 1). The fractions of overlapping gene pairs and leading genes do not add up to 100% because all non-overlapping genes with <200 nt between the given gene and the upstream one are not included in any of these groups.

Next, we estimated the fractions of overlapping genes with a SD motif located in the 3′ region of the upstream gene, which suggests termination-reinitiation. The results for all 720 species are included in Supplementary Data 1, and the mean fractions for the 24 groups of prokaryotes are shown in Fig. 3a. In 20 of the 24

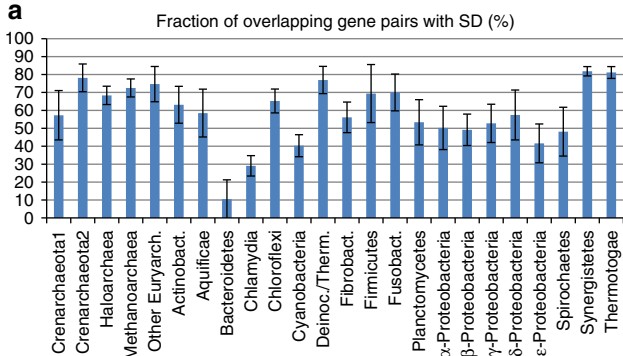

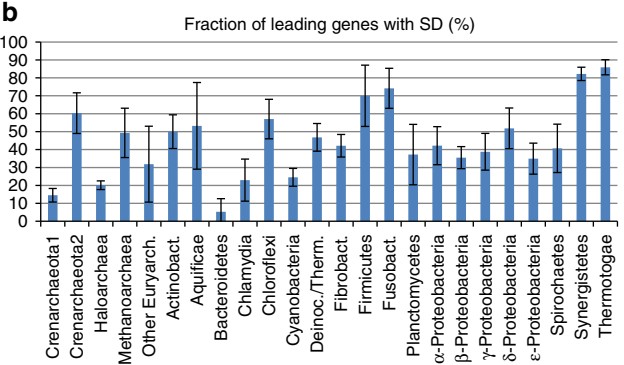

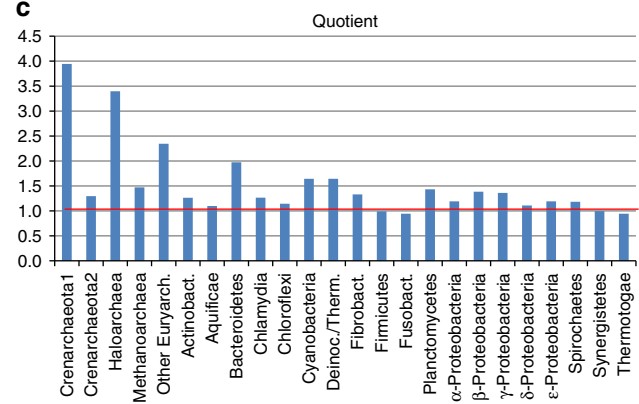

**Fig. 3** SD sequences in 720 genomes of 24 groups of prokaryotes. **a** Fractions of overlapping gene pairs that contain a SD motif upstream of the downstream gene, in the 3′-region of the upstream gene. Mean values and standard deviations are shown. **b** Fractions of leading genes that are preceded by a SD motif. Mean values and standard deviations are shown. **c** The quotient of the values shown in **a** and **b**, which represents the relative importance of the SD motif at downstream genes of overlapping gene pairs versus leading genes

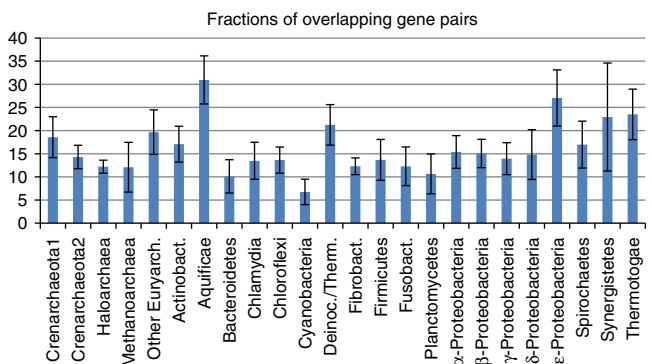

**Fig. 2** Overlapping gene pairs in 720 genomes of 24 groups of prokaryotes. Fractions of overlapping gene pairs (%). Mean values and standard deviations are shown

groups, the fractions are above or around 50%, and in 10 groups, the values are higher or around 70%. Thus, when genes overlap, the start codon of the downstream gene is typically preceded by a SD motif in most archaea and bacteria. Only three groups have a lower fraction of SD, namely, *Bacteroidetes* with 10%, *Chlamydia* with 30%, and *Cyanobacteria* with 40%. Therefore, in these groups, the SD motif is typically not needed for translation (re) initiation, irrespective of the existence of translational coupling.

For comparison, we estimated the fractions of the leading genes that are preceded by a SD motif. Figure 3b shows that the values for the leading genes are much more variable than those for the overlapping genes. Only four groups have mean values of 70% or higher (*Firmicutes*, *Fusobacteria*, *Synergistetes*, and *Thermotogae*), showing that SD is typically involved in de novo initiation at 5′-UTRs. Five groups have fractions of about 20% or less, i.e. *Haloarchaea*, *Crenarchaeota I* (including e.g. *Acidianus* and *Sulfolobus*), *Bacteroidetes*, *Chlamydia*, and *Cyanobacteria*.

To predict the relative importance of the SD motif for reinitiation in overlapping gene pairs and for de novo initiation at leading genes, the ratios (quotients) of the above-mentioned values were calculated (Fig. 3c). For 18 of the 24 groups, the quotient value is around 1, suggesting about equal importance of the SD motif for reinitiation at overlapping genes as well as for de novo initiation at 5′-UTRs. Notably, no group has a quotient considerably <1.0. In contrast, four groups show a value of 2.0 or higher, suggesting that the SD motif has a higher importance for reinitiation than for de novo initiation.

Haloarchaea and Bacteroidetes do not use the SD motif for de novo initiation[26,27] and their associated quotient values, 3.5 and 2.0, respectively, appear to be lower than expected (Fig. 3c). Over-predictions of the fractions of SD motifs could potentially account for this discrepancy. We calculated the fractions of "strong" SD motifs, with 16S rRNA binding energies <−8.4 kcal/mol[28]. For overlapping gene pairs, the strong SD motifs represented a relatively small fraction of all identified instances of SD (Supplementary Fig. 2A), enabling a better differentiation between the groups. The decrease was even more pronounced for leading genes (Supplementary Fig. 2B). The values for *Haloarchaea* and *Bacteriodetes* are close to zero, in better agreement with the expectations. To address the relative importance of strong SD motifs for reinitiation at overlapping genes and de novo initiation, we estimated the quotients of the two fractions (Supplementary Fig. 2C). Six groups had quotients below 1.0, suggestive of a higher importance of strong SD motifs for de novo initiation than for reinitiation. In contrast, for 12 of the 24 groups, the values were greater than 2.0, which is compatible with a greater importance of strong SD motifs for reinitiation than for de novo initiation.

**Termination-reinitiation at overlapping genes in *H. volcanii*.** The bioinformatics analyses showed that all groups of prokaryotes contain substantial fractions of overlapping gene pairs. We chose two model organisms to determine experimentally whether or not (strict) translational coupling is typical of overlapping gene pairs in prokaryotes. The first species was *H. volcanii*, a model species of the haloarchaea, and the second species was *E. coli*, the most intensely studied bacterium (see below).

Four gene pairs were selected from the *H. volcanii* genome that had the most common overlap of four nucleotides (**AUGA**) and strong SD motifs of five or six nucleotides. The gene pairs encoded two subunits of the enzyme urease, an amino acid binding protein/homoserine dehydrogenase, and two pairs of ribosomal proteins (Supplementary Table 1). The upstream genes and the 5′-portions of the downstream genes were cloned into an expression vector for *H. volcanii*. The downstream genes were

cloned as translational fusions to the *dhfr* (dihydrofolate reductase) gene, which is routinely used as a reporter gene for *H. volcanii*[27,29,30]. In addition, mutated versions were generated that contained a stop codon in the 5′-part of the respective upstream gene (Fig. 4a). This premature stop prevented ribosomes translating the upstream genes from reaching the gene overlaps. In all cases, the DHFR specific activities were quantified using an enzymatic assay (Supplementary Fig. 3A), and the *dhfr* transcript levels were quantified by Northern blotting (Supplementary Fig. 3B). These values were used to calculate translational efficiencies, which in all cases were normalized to the version lacking the stop codon (Fig. 4b). In all four cases, the translational efficiencies of the stop codon-containing versions were close to zero, showing that strict translational coupling occurred at the four gene pairs and translation of the downstream genes absolutely depended on the translation of the upstream gene. Notably, de novo initiation at the SD motif of the downstream gene did not occur, in agreement with the fact that, in Haloarchaea, the SD motif is non-functional in de novo translation initiation in 5′-UTRs[27]. To extend this analysis, one additional gene pair was chosen, which contained no overlap, but rather, a short intergenic region of 10 nt (*flaA1-flaA2*) (Supplementary Table 1). Similarly to the overlapping gene pairs, also in this case, quantification of the translational efficiencies showed that translation of the downstream gene strictly depended on the translation of the upstream gene (Fig. 4b).

Next, we aimed to clarify whether the 5'-portions of the downstream genes in overlapping gene pairs are required for translational coupling, or whether coupling is solely encoded in the upstream gene. To this end, four additional gene pairs were chosen and the upstream genes were directly fused to the *dhfr* reporter gene (Fig. 4c). Two gene pairs contained a four nucleotide overlap, one pair contained a one nucleotide overlap, and another one contained an intergenic region of one nucleotide (Supplementary Table 1). Again, mutagenesis was used to generate stop codon-containing variants, and the levels of the *dhfr* transcript and DHFR protein as well as the translational efficiencies were quantified for all 8 variants (Supplementary Fig. 3, Fig. 4d). In all four gene pairs, the downstream *dhfr* gene was not translated when ribosomes were prevented by a premature stop codon from reaching the overlap. These results showed that the sequences of the upstream genes were sufficient for coupling to occur. Together, these findings validate strict translational coupling for 9 out of 9 analyzed gene pairs of *H. volcanii*. To our knowledge, these results represent the first experimental demonstration of translational coupling in archaea.

**Termination-reinitiation at overlapping genes in *E. coli*.** Translational coupling in *E. coli* has been, to some extent, characterized in the past. However, typically, in such experiments, viral or eukaryotic genes were used and the transcript levels were not analyzed. Because transcript stability in *E. coli* is dramatically affected by the efficiency of translation[31–33], it remains unclear whether these previous studies really quantified translational coupling, or transcript stabilities, or a mixture of both. Therefore, we decided to quantify translational coupling at gene pairs in *E. coli*, with specific consideration of the following: (1) native cellular gene pairs were studied, (2) gene pairs with the most common four nucleotide overlap were used, (3) both transcript levels and protein levels were quantified, and (4) SD motifs of different strengths were chosen.

Five gene pairs were chosen for experimental analysis (Supplementary Table 1). The gene pairs encoded two subunits of a hydrogenase, two subunits of a dehydrogenase, two subunits of biotin synthase, two hypothetical proteins, and two subunits of

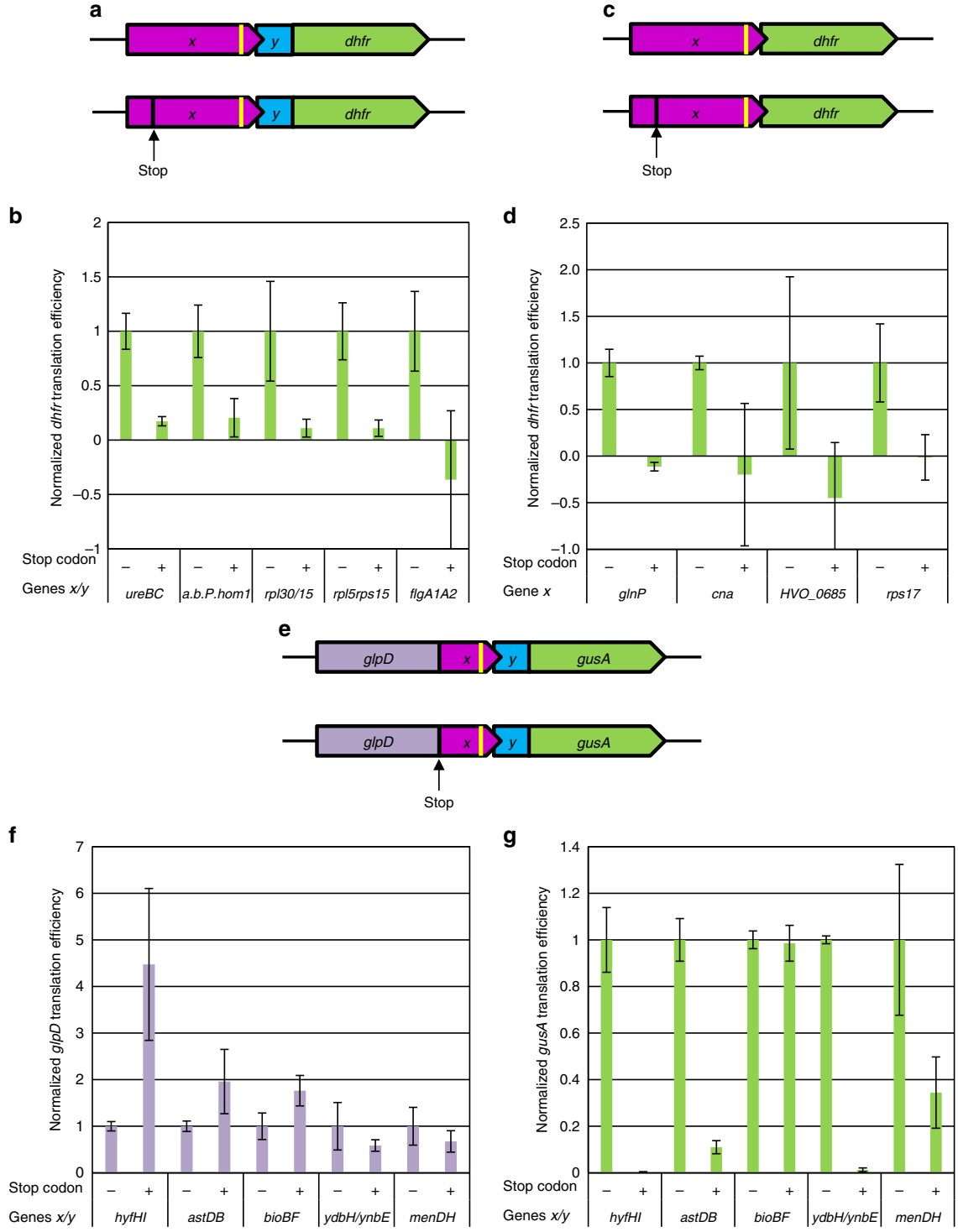

a menachinone biosynthesis enzyme. The overlapping regions of these five gene pairs were cloned into a dual reporter gene vector (Fig. 4e). As a result, translational fusions were generated of the *glpD* reporter gene with the upstream gene and of the downstream gene with the reporter gene *gusA*. Also, for the *E. coli* gene pairs, stop codon-containing and stop codon-lacking variants were generated and compared. Initially, the entire ORFs of the gene pairs were used. However, it turned out that the stop codon-containing transcripts were strongly destabilized such that full-length bicistronic transcripts could not be detected at all, preventing quantification of translational efficiencies. Therefore, the cloned regions were trimmed to the last 99 nt of the upstream

genes and the first 30 nt of the downstream genes. This arrangement stabilized the transcripts, while still guaranteeing that the ribosomes were bound exclusively to native gene sequences during the presumed termination and reinitiation. For all 10 constructs, the GlpD and GusA protein levels were quantified using enzymatic assays, and the levels of the bicistronic transcripts were quantified using Northern blotting with *glpD*- and *gusA*-specific probes (Supplementary Fig. 4). For the *hyfHI* gene pair, introduction of a stop codon considerably destabilized the bicistronic transcript, whereas, in contrast, it stabilized the bicistronic transcript for the *menDH* gene pair. For the other three gene pairs, the transcript levels for the two variants were

**Fig. 4** Translational coupling via the TeRe mechanism in *H. volcanii* and *E. coli*. **a** Schematic overview of the constructs for *H. volcanii*. The native overlapping gene pairs are shown in magenta and blue, the *dhfr* reporter gene is shown in green. The intragenic SD motif is indicated by a yellow bar. The designed premature stop codon is indicated. **b** Normalized translational efficiencies of the native gene pairs lacking a premature stop codon (-) and the stop codon variants (+). The names of the gene pairs are shown at the bottom. Average results of three biological replicates and their standard deviations are shown. Source data are provided as Source Data file. **c** Schematic over view of the constructs for *H. volcanii*. The native upstream gene is shown in magenta, the *dhfr* reporter gene is shown in green. The intragenic SD motif is indicated by a yellow bar. The designed premature stop codon is indicated. **d** Normalized translational efficiencies of the native genes lacking a premature stop codon (−) and the stop codon variants (+). The names of the genes are shown at the bottom. Average results of three biological replicates and their standard deviations are shown. Source data are provided as Source Data file. **e** Schematic overview of the constructs for *E. coli*. The native overlapping gene pairs are shown in magenta and blue, the glpD reporter gene is shown in purple, the *dhfr* reporter gene is shown in green. The intragenic SD motif is indicated by a yellow bar. The presence of a stop codon is indicated. **f** Normalized *glpD* translational efficiencies of the native gene pairs lacking a stop codon (−) and the stop codon variants (+). The names of the gene pairs are shown at the bottom. Average results of three biological replicates and their standard deviations are shown. Source data are provided as Source Data file. **g** Normalized *gusA* translational efficiencies of the native gene pairs lacking a stop codon (−) and the stop codon variants (+). The names of the gene pairs are shown at the bottom. Average results of three biological replicates and their standard deviations are shown. Source data are provided as Source Data file

closely similar. These results underscore the importance of the transcript level analyses when translation efficiency is measured.

The protein and transcript levels were used to calculate translational efficiencies (Fig. 4f, g). For all gene pairs, the GlpD levels in the stop codon-containing variants were at least as high as those in the stop codon-lacking variants. Surprisingly, in the case of *hyfHI*, introduction of a stop codon increased the GlpD amount more than fourfold. The most likely explanation is that the C-terminus of HyfH reduced the specific activity of GlpD in the fusion protein.

Quantification of the *gusA* translational efficiencies did not yield uniform results. In three cases, translation of the downstream gene in the stop codon-containing variants was extremely low or undetectable, indicating strong translational coupling for *hyfHI*, *astDB*, and *ydbH/ynbE*. For the *menDH* gene pair translational coupling appeared to account for about 60% of the initiation of the downstream gene, whereas nearly 40% depended on de novo initiation. Translational coupling seemed not to occur in the *bioBF* gene couple, but the full translational efficiency was observed when the ribosomes did not reach the overlap in the stop codon-containing variant. This result was particularly surprising as the *bioF* gene is preceded by a very weak SD motif that would not be expected to mediate efficient de novo initiation (Supplementary Table 1). In summary, with one exception, the analyzed gene pairs from *E. coli* showed translational coupling in the double reporter gene system.

**The role of Shine-Dalgarno motifs in termination-reinitiation.** Our next goal was to analyze whether the intragenic SD motifs in the upstream genes are important or even essential for the termination-reinitiation in *H. volcanii* and/or in *E. coli*. To this end, the SD motif was eliminated from 7 gene pairs of *H. volcanii* (Supplementary Table 2, and Fig. 5a, c). As described above, the DHFR protein levels and the *dhfr* transcript levels were quantified (Supplementary Fig. 5). The transcript levels were nearly uniform, with one exception. Removal of the SD motif from *rpl15* led to a more than threefold increase in the transcript level, again underscoring the importance of transcript analyses. The results were used to calculate translational efficiencies (Fig. 5b, d). In none of the cases, translation of the downstream gene was eliminated, showing that the SD motif is not essential for termination-reinitiation in *H. volcanii*. Reinitiation efficiencies were found to be highly variable and depended on the individual gene pair. In two cases, the efficiency of reinitiation in the absence of the SD motif was reduced by about 80%, indicating high importance of the SD motif (*glnP* and *h.p.*). In three cases, the reinitiation efficiencies were reduced by about 50%, and in one case by only 20%. No correlation was found between the substantially different contributions of the SD motif and features

such as SD motif length or distance between the SD motif and start codon. For example, *glnP* and *cna* both contain a 5 nt SD motif and have an identical distance between SD motif and start codon, yet the efficiencies of reinitiation varied greatly (Fig. 5d). Surprisingly, in one of the 7 cases, the *rpl30/rpl15* gene pair, the mutation of the SD motif led to a considerable increase in reinitiation efficiency (Fig. 5b). It seems that the contribution of the SD motif for the efficiency of termination-reinitiation in *H. volcanii* is highly variable, suggesting that additional sequences, possibly, those around the SD motif, substantially affect translational coupling. These results also emphasize the importance of analyzing multiple pairs of native genes for understanding the mechanisms of translational coupling.

Next, the importance of the SD motif for termination-reinitiation in *E. coli* was analyzed by generation of SD-less mutants of the 5 gene pairs (Fig. 5e, Supplementary Table 2). The transcript and protein levels for native genes and the SD-less mutants were quantified (Supplementary Fig. 6). The transcript levels were closely similar, with one exception. Removal of the SD motif from *menDH* led to a more than twofold increase of the transcript level. This shows how large the impact of just six nucleotides can be on the stability of a large transcript of about 2000 nt. The results of the quantifications of protein and transcript were used to calculate the translational efficiencies (Fig. 5f, g). For one gene pair (*ydbH/ydbE*), the SD motif turned out to be strictly essential for reinitiation. In three other cases, removal of the SD reduced the efficiency of reinitiation by 60–90%, showing the high importance of the SD motif., Removal of the SD motif from the *bioBF* gene pair had no influence on reinitiation. However, *bioBF* is a unique case because the analysis of the stop codon variant showed that, at least in the analyzed construct, no translational coupling occurred, but instead, translation of *bioF* initiated de novo (Fig. 4g).

Taken together, similarly to the case of *H. volcanii*, the results obtained with 5 *E. coli* gene pairs also showed that the contribution of the SD motif to termination-reinitiation depends on the individual gene pair and can vary from strictly essential to none. Nevertheless, in 4 of the 5 analyzed *E. coli* gene pairs, removal of the SD motif substantially reduced the efficiency of reinitiation.

**The putative mechanism of TeRe in *E. coli*.** In *H. volcanii*, the SD motif does not operate at 5′-UTRs, and therefore, there is no reason to believe that it would mediate de novo initiation at overlapping gene pairs. In contrast, in *E. coli*, the SD motif mediates de novo initiation at the 5′-UTRs, and thus, de novo initiation can be expected to occur at intragenic SD motifs at gene overlaps as well. Therefore, de novo initiation has to be prevented at these sites to enable translational coupling. We hypothesized

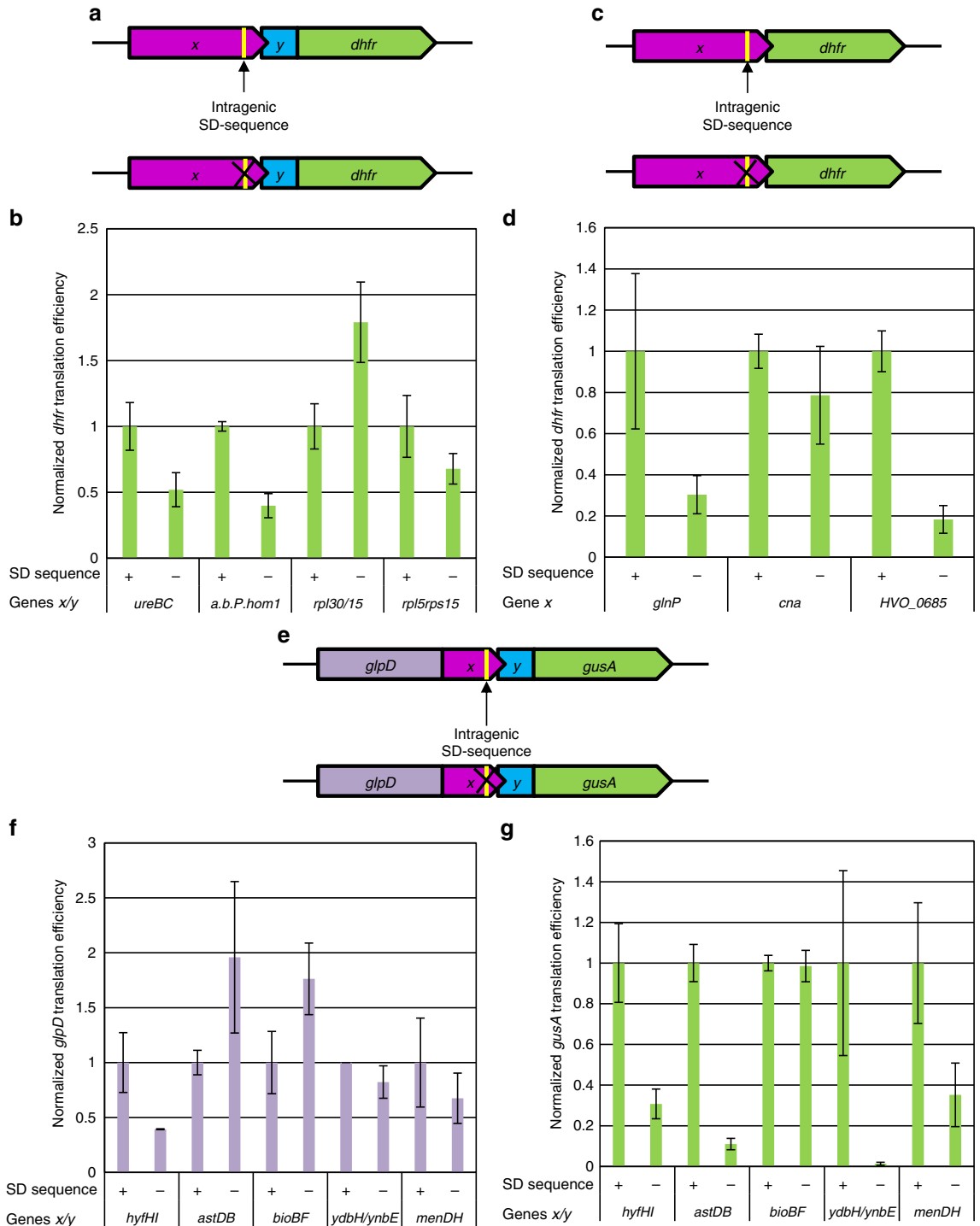

**Fig. 5** The importance of the SD motif for termination-reinitiation. **a** Schematic overview of the constructs for *H. volcanii*. The native overlapping gene pairs are shown in magenta and blue, the *dhfr* reporter gene is shown in green. The intragenic SD motif is indicated by a yellow bar, which is crossed in the SD mutants. **b** Normalized translational efficiencies of the native gene pairs containing (+) or lacking (−) a SD motif. The names of the gene pairs are shown at the bottom. Average results of three biological replicates and their standard deviations are shown. Source data are provided as Source Data file. **c** Schematic overview of the constructs for *H. volcanii*. The native upstream gene is shown in magenta, the *dhfr* reporter gene is shown in green. The intragenic SD motif is indicated by a yellow bar, which is crossed in the SD mutants. **d** Normalized translational efficiencies of the containing (+) or lacking (−) a SD motif. The names of the genes are shown at the bottom. Average results of three biological replicates and their standard deviations are shown. Source data are provided as Source Data file. **e** Schematic overview of the constructs for *E. coli*. The native overlapping gene pairs are shown in magenta and blue, the glpD reporter gene is shown in purple, the *dhfr* reporter gene is shown in green. The intragenic SD motif is indicated by a yellow bar, which is crossed in the SD mutants. **f** Normalized glpD translational efficiencies of the native gene pairs with (+) or without (−) a SD motif. The names of the gene pairs are shown at the bottom. Average results of three biological replicates and their standard deviations are shown. Source data are provided as Source Data file. **g** Normalized gusA translational efficiencies of the native gene pairs with (+) or without (−) a SD motif. The names of the gene pairs are shown at the bottom. Average results of three biological replicates and their standard deviations are shown. Source data are provided as Source Data file

that local RNA structures might prevent de novo initiation at downstream genes, and to test this hypothesis, the cloned regions of the 5 gene pairs were folded in silico (Supplementary Fig. 7). For *hyfHI*, *astDB*, and *ydbH/ynbE*, stable double-stranded structures were predicted at or very close to the TIR, which could explain the lack of do novo initiation at the respective downstream genes (Supplementary Fig. 7). The predicted structure for *bioBF* does not include extended double-stranded regions around the TIR (Supplementary Fig. 7), which is in agreement with the experimentally observed de novo initiation. The predicted structure for *menDH* includes double-stranded regions of intermediate length (Supplementary Fig. 7), which might explain the partial inhibition. In summary, the predicted structures of the mRNAs around the overlaps of the five gene pairs are all in congruence with the experimental results.

## Discussion

The bioinformatics analyses of 720 genomes of archaea and bacteria showed that all groups of prokaryotes contain considerable fractions of overlapping co-directional gene pairs. If gene overlaps are taken as a prediction of translational coupling, these results suggest that translational coupling could be widespread in prokaryotes, although so far it has been experimentally studied only in very few species. One gene pair of the archaeon *Sulfolobus solfataricus* has been analyzed using an in vitro translation assay[34]. The genes had adjacent stop and start codons (UGAGUG) and a SD motif within the ORF of the upstream gene. Analysis of mutants has shown that, in this case, translation of the two genes was not coupled, in spite of the close spacing without an intergenic region[34]. Clearly, experimental data on more species are needed to clarify in which groups of prokaryotes gene overlaps are indicative of translational coupling.

Importantly, not all gene pairs in polycistronic transcripts overlap. Instead, many gene pairs are separated by intergenic regions, and thus, the genes are either translated independently or are translationally coupled via the UTNI mechanism, which involves de novo initiation at the downstream gene (Fig. 1a, b). For three operons, which encode ribosomal proteins, the F-type ATPase, and Cas proteins, it has indeed been shown that the protein levels of downstream genes can exceed those of the upstream genes several fold[6].

In addition to the fraction of overlapping genes, the fractions of genes that are preceded by a (strong) SD motif were determined for overlapping gene pairs (reinitiation) and for leading genes (de novo initiation). The results were not uniform for the 24 analyzed groups of prokaryotes, and four distinct patterns of SD usage were observed (see Fig. 3 and Supplementary Fig. 2).

The first pattern combines groups of prokaryotes that make intermediate use of the SD motif for de novo initiation as well as for reinitiation and have very low fractions of strong SD motifs. A variety of groups of bacteria show this pattern, including the *Fibrobacteria*, all five groups of proteobacteria, the *Planctomycetes*, and the *Spirochaetes*. Therefore, the results that were obtained with *E. coli* seem to be typical of all γ-proteobacteria and beyond.

The second pattern is used by five bacterial phyla, *i.e. Aquificae*, *Firmicutes*, *Fusobacteria*, *Synergistetes*, and *Thermotogae*. It is characterized by a high usage of the SD motif as well as the strong SD motif for de novo initiation as well as for reinitiation (Supplementary Fig. 2B).

The third pattern is characteristic of one group of *Crenarchaeota* (*Crenarchaeota 1*), *Haloarchaea*, and the *Cyanobacteria*. These groups make only infrequent use of the SD motif for de novo initiation, if they use it at all. One reason for the infrequent use or lack of use of SD motifs for de novo initiation in the two

groups of Archaea might be the high fraction of leaderless transcripts, which do not contain 5′-UTRs and thus cannot harbor a SD motif. The fractions of leaderless transcripts are 69% for *Sulfolobus solfataricus*[35] and 72% for *H. volcanii*[36], two representatives of *Crenarchaeota 1* and *Haloarchaea*. In a stark contrast to the lack of importance for de novo initiation, *Crenarchaeota I, Haloarchaea*, and *Cyanobacteria* make ample use of the SD motif for reinitiation at overlapping gene pairs.

The fourth pattern was found only in *Bacteroidetes*. This group does not seem to make use of the SD motif at all, neither for de novo initiation nor for reinitiation, because the fractions are below 10%.

Overall, the usage of the (strong) SD motif for de novo initiation at 5′-UTRs and for reinitiation at overlapping genes is highly variable in different groups of archaea and bacteria. Notably, in 3 groups of archaea and 3 groups of bacteria, the SD motif seems to be more important for reinitiation than for de novo initiation (values of >1.5 in Fig. 3c).

Transcripts that contain long non-translated parts turned out to be stable in *H. volcanii*, in contrast to *E. coli*. This enabled the usage of mutated, stop codon-containing whole ORFs of nine upstream genes for testing the prediction that translational-coupling is typical for overlapping genes in haloarchaea (Fig. 4a–d). In all cases premature termination in the upstream ORF led to a total lack of translation of the downstream gene, showing the strict dependence of the nine downstream genes on translational coupling to the upstream genes. It has been shown that SD motifs are not functional in attracting 30 S subunits for de novo initiation at 5′-UTRs in *H. volcanii*[27]. Therefore, there is no reason to believe that de novo initiation can occur at downstream genes of operons. Thus it seems likely that translational coupling occurred via the termination-reinitiation and not via the UTNI mechanism. The high number of 433 gene pairs with a 1-nt or 4-nt overlap[27] indicates that termination-reinitiation operates at more than 10% of all *H. volcanii* genes and thus is essential for a considerable fraction of the proteome. Notably, this is the first experimental evidence for translational coupling in any archaeal species.

A variety of experimental studies have addressed translational coupling in *E. coli*. However, many of these included analysis of phage genes[19,37,38] or eukaryotic genes[25], and/or did not take the transcript levels into account[15,39,40].

In contrast to these previous studies, we characterized the patterns of translation in five native, overlapping *E. coli* gene pairs for which we quantified the transcript and protein levels. Notably, the short region from −99 to +30 nt was sufficient to ensure translational coupling in 4 of the 5 analyzed examples (*hyfHI*, *astDB*, *ydbH/ynbE*, and *menDH*), ruling out a major role of long-range interactions and strongly indicating that coupling occurred via termination-reinitiation. Only in one of the 5 cases, the *bioBF* gene pair, translational coupling was not observed. In this case, coupling via the UTNI mechanism might require long-range interactions that are absent in the cloned fragment, or no translational coupling occurs at all.

The in silico folding of the cloned fragments indicated the possibility of the formation of local structures around the start/stop overlap (with the exception of *bioBF*), which would explain the lack of translation of the downstream genes in spite of the presence of SD motifs. These structures would only be melted when the ribosome that translates the upstream gene reaches the termination region, excluding other ribosomes from initiation at the downstream gene. It remains to be clarified whether the large subunit is exchanged during termination-reinitiation so that only the 30S subunit remains bound to the mRNA (like in eukaryotic viruses[17]), or whether the 70S ribosome mediates termination-reinitiation.

It should be noted that translational coupling and de novo initiation are not mutually exclusive in *E. coli*. An elegant study on a synthetic bicistronic operon with a 50-bp intergenic region has shown[41] that de novo initiation at the downstream gene occurs, as expected, but in addition, translation of the upstream gene enhances translation of the downstream gene. An increase of the translational efficiency of the upstream gene by a factor of 200 led to a fivefold increase in the translation of the downstream gene, demonstrating partial coupling of the two genes despite the long intergenic region. Thus, mixed mechanisms of translation reinitiation appear to operate in *E. coli*, in addition to the three distinct mechanisms of de novo initiation and translational coupling (Fig. 1).

In overlapping genes, the SD motif is located within the open reading frame of the upstream gene, putting an extra sequence constraint on the C-terminus of the encoded protein, which clearly incurs a fitness cost. Nevertheless, genomes of most groups of prokaryotes contain high fractions of overlapping genes. Therefore, there must be evolutionary advantages that compensate for the disadvantage of the additional protein sequence constraint. Experimental characterization of 14 gene pairs has demonstrated tight translational coupling in both a model archaeon and a model bacterium, suggesting that translational coupling has probably driven the evolution of gene overlaps.

One evolutionary advantage of termination-reinitiation might be the production of exactly equal levels of the two proteins encoded by the overlapping gene pairs. This could be highly beneficial or even essential for the cell, for example, when: (1) the two genes encode a catalytic and a regulatory subunit of an enzyme, and the production of the catalytic subunit alone would result in useless dissipation of energy, (2) one or both of the proteins is unstable as a monomer and thus degradation is inhibited by heterodimer formation, (3) one or both of the proteins is toxic to the cell in its monomeric form, or (4) one protein require the other as a guide to reach the correct intracellular localization. However, production of equal amounts of two proteins encoded by overlapping gene pairs would require extremely efficient coupling efficiency of 100%, and thus, in future work, it will be important to quantify the coupling efficiency in diverse bacteria and archaea. Nevertheless, despite the limited scope of the present study, the combination of bioinformatics and experimental analyses clearly demonstrates the wide spread of overlapping gene pairs in archaea and bacteria, and the importance of gene overlaps for translational coupling via termination-reinitiation.

## Methods

**Archaeal and bacterial strains, media, and growth conditions**. *H. volcanii* strain H26 was obtained from Thorsten Allers[42] (University of Nottingham, UK). The strain contains a *pyrE2* deletion and is thus auxotrophic for uracil. The *dhfr* gene (HVO_1279) has been deleted in the genome to enable the usage of plasmid-bound copy as a reporter gene[43].

*H. volcanii* was grown in complex medium containing 0.3% (w/v) yeast extract and 0.5% (w/v) tryptone[43]. The medium also contained 2.1 M NaCl, 220 mM MgCl$_2$, 41 mM MgSO$_4$, 13 mM KCl, 9 mM CaCl$_2$, and 50 mM Tris/HCl, pH 7.2. The medium was complemented with 8 µM FeSO$_4$, 0.1% (v/v) SL-6 trace element solution, 0.1% (v/v) vitamin solution (B6891, Sigma-Aldrich, St. Louis, MO, USA), and 50 µg/ml uracil.

The *E. coli* strain JW3389-1 was obtained from the Keio collection[44] (http://cgsc.biology.yale.edu/Keiolist.php). It has a deletion of the chromosomal copy of the *glpD* gene and thus allows its usage as a reporter gene. To enable the simultaneous use of the *gusA* gene as a reporter gene, the chromosomal *gusA* gene has also been deleted[33].

*E. coli* was grown in SOB complex medium containing 2% (w/v) tryptone and 0.5% (w/v) yeast extract[45]. The medium also contained 10 mM NaCl, 2.5 mM KCl, 10 mM MgCl$_2$, and 10 mM MgSO$_4$.

**Construction of reporter gene plasmids**. General molecular genetic techniques were performed as described by Green and Sambrook[46]. The backbone of the *H.*

*volcanii* vector pTA929[47] was used for expression of the fusion genes. The vector contains replication origins and selection genes for *E. coli* as well as for *H. volcanii*. First, the *dhfr* gene was amplified by PCR and cloned into the vector pTA929 using the restriction sites *NcoI* and *KpnI*, yielding the plasmid pBJ2. The oligonucleotides that were used to amplify the *H. volcanii* genes are listed in Supplementary Table 3. The fragments either contained the upstream gene and 30 nt of the downstream gene, or they contained the upstream gene (compare Fig. 5). The fragments were cloned into pBJ2 using the restriction sites *NdeI* and *NcoI*.

For expression of the *E. coli* fusion genes the backbone of the plasmid pSK+ was used (Stratagene). The *glpD* gene and the *gusA* gene had been cloned and used as reporter genes previously[33]. The oligonucleotides that were used to amplify fragments from overlapping gene pairs of *E. coli* are also listed in Supplementary Table 3. The fragments contained 99 nt of the upstream gene and 30 nt of the downstream gene. They were cloned into the vector using the restriction sites *NcoI* and *XhoI*.

The oligonucleotides that were used to exchange the SD motifs with unrelated sequences are shown in Supplementary Table 4. For the mutagenesis the cloned gene pairs were subcloned from the shuttle plasmids into the vector pSK+ to reduce the plasmid size. The quickchange site directed mutagenesis kit was used to introduce the mutations (www.agilent.com). Then the altered fragments were cloned back into the shuttle vector. After all PCR, cloning and mutagenesis steps the DNA sequences were verified.

**Quantification of specific reporter enzyme activities**. The activity of the DHFR reporter enzyme was quantified with an enzymatic assay[30]. In short, the oxidation of NADPH by dihydrofolic acid was followed at 340 nm, and an extinction coefficient of 6.22 mM$^{-1}$ cm$^{-1}$ was used to calculate the activity. The assay contained 3 M KCl, 50 mM K$_2$HPO$_4$, 50 mM KH$_2$PO$_4$, 50 mM citrate, 2 mM NADPH, and 0.6 mM dihydrofolate. The assays were performed in a volume of 250 µl in microtiter plates at 42 °C. The protein concentrations were quantified using the BCA assay (www.thermofischer.com) with BSA as standard and used to calculate the specific activities (nkat/mg). A detailed description of the method has been submitted to the Nature Protocol Exchange database (https://doi.org/10.21203/rs.2.11263/v1).

The GlpD and GusA reporter gene activities were quantified with enzymatic assays[33]. In short, the GlpD assay made use of the artificial redox mediator PMS and the artificial electron acceptor MTT. Product formation was followed at 570 nm and quantified using an extinction coefficient of $1.7 \times 10^4$ M$^{-1}$ cm$^{-1}$. The assay contained 1.125 mM MTT, 5.63 mM PMS, 0.4% Triton X-100, 3.81 mM DL-glycerol-3-phosphate, and 24 mM Tris/HCl pH 7.2. The GusA activity was quantified with pNPG (para-nitrophenyl-glucoronic acid) as substrate. Formation of the product para-nitrophenol was followed at 405 nm and quantified using an extinction coefficient of $2.18 \times 10^5$ M$^{-1}$ cm$^{-1}$. The assay contained 100 mM sodium phosphate pH 7.0, 20 mM beta-mercapto ethanol, 0.2% Triton X-100, and 0.8 mg/mg para-nitrophenylphosphate. Also the assays of the two *E. coli* enzymes were performed in microtiter plates at 37 °C. Protein concentrations were determined as described above, and specific activities were calculated. Detailed description of the methods have been submitted to the Nature Protocol Exchange database (GlpD: https://doi.org/10.21203/rs.2.10595/v1; GusA: https://doi.org/10.21203/rs.2.10596/v1).

For all assays cultures of the mid-exponential growth phase ($4-5 \times 10^8$ cells/ml) were used. In each case three biological replicates were performed, and average values and standard deviations were calculated.

**Quantification of transcript levels**. The samples for RNA isolation were taken from the same cultures simultaneously with the samples for the reporter gene assays described above. RNA was isolated using the RNeasy Mini kit (Qiagen, Hilden, Germany). Relative transcript levels were quantified by Northern blot analyses[33]. In short, RNA was fractionated on agarose gels and transferred to nylon membranes by downward blotting. Probes were generated by PCR and labeled with DIG-dUTP. The oligonucleotides for probe generation are summarized in Supplementary Table 5. After hybridization, an enzyme-coupled anti-DIG antibody and the chemo luminescence substrate CDP-Star (Roche, Mannheim, Germany) were used to assay the probes, and X-ray films were used to detect the emitted light. The films were scanned and the signals were quantified using the software ImageJ (http://rsbweb.nih.gov/ij). The signals were normalized to the amounts of the 16S rRNA. Three biological replicates were performed, and average values and standard deviations were calculated. A detailed description of the method has been submitted to the Nature Protocol Exchange database (https://doi.org/10.21203/rs.2.11264/v1).

**Calculation of normalized translational efficiencies**. Translational efficiencies were calculated as quotients of the specific enzyme activities and the relative transcript levels that were normalized to a control, as indicated. A strain with an empty vector was used as a negative control and its "translational efficiency" was subtracted from that of all samples. When the translational efficiencies of the samples are zero, subtraction of the negative control results in very small values that, due to statistical variations, can be positive or negative.

For a better comparative visualization of the effects of a premature stop codon or the replacement of the SD motif, the results of the mutants were normalized to that of the controls.

A detailed description of the method has been submitted to the Nature Protocol Exchange database (https://doi.org/10.21203/rs.2.11363/v1).

**Computational genome analyses**. To analyze the potential cases of translation reinitiation in prokaryotes, we selected completely sequenced genomes from 720 representative species of bacteria and archaea (see Supplementary Data1 for the list of the analyzed genomes with the accession numbers).

The computational pipeline employed for the identification of clusters of homologous proteins is illustrated in Supplementary Fig. 8. Protein sequences extracted from a database containing all complete prokaryote genomes available in 2014 were clustered using the USEARCH software (version 8.1)[48] with a sequence identity threshold of 90%. The resulting groups (hereafter Tight Clusters) contained protein sequences that shared at least 90% sequence identity. For each of the Tight Clusters, we selected representative proteins (TCrep) composed of all protein sequences from the dataset of the 720 representative genomes (hereafter rep720) or the longest protein sequences in the Tight Cluster when no proteins from rep720 were present in a Tight Cluster. This set of representative protein sequences was clustered into Loose Clusters with USEACRH[48] using a threshold of 50% sequence identity. Loose Clusters that did not include protein sequences from rep720 were discarded.

The Loose clusters containing more than 10 proteins (hereafter Large Loose Cluster, LLC) were converted into BLAST databases[49]. In order to validate the clustering results, proteins from rep720 were identified in each LLCand compared to the rest of the sequences from the same LLC using BLASTp[49]. The 10 best BLAST hits with $e$-value ≤10$^{-4}$ (no low complexity filtering, default composition-based statistics adjustment) were selected as the set of homologs for the given rep720 protein query.

The loose clusters containing 10 or less protein (Small Loose Cluster, SLC) were validated using a different procedure. For each SLC, proteins from rep720 were extracted and compared to the TCrep sequence set (representative sequences extracted from Tight Clusters). Sets of homologs were selected by the same criteria as for the LLC.

As a result of these procedures, each protein sequence from rep720 becomes associated with zero, one, or a set of homologs from either Tight or Loose clusters. In order to generate reliable, uniform sets of homologs, only the proteins with more than 10 homologs were kept, 10 best non-identical hits were selected.

Identification and analysis of overlapping genes and intergenic distances critically depend on the correct annotation of both translation start and stop codon positions. Because stop codon readthrough is relatively rare, annotation of the translation stop is generally straightforward. In contrast, the codons used to start translation also encode regular amino acids, making the annotation of the correct translation initiation site (TIS) challenging. However, location of the start codon is generally well conserved in evolution, largely, due to the constraints on the upstream regions that are required to initiate translation. Using alignments of homologous genes, we evaluated the conservation of the start position and re-annotated the start codons in cases when the currently annotated start was not conserved but an alternative, conserved start codon was identified.

Start positions were evaluated only for protein-coding sequences from rep720 for which sets of homologs were identified (see the preceding section and Supplementary Fig. 6). Each protein sequence was aligned with the selected set of 10 homologous protein sequences using MUSCLE[50]. The alignment was scanned with a sliding window of 3 positions for which the number of potential starts (i.e. the first residue of the respective protein) in the homologous protein sequences was counted. Windows including at least five potential starts among the homologous sequences were selected for further analysis. The consensus start position was set as the alignment position associated with the highest protein start position (among the homologs) and the closest to the annotated start position of the query.

To re-annotate the start of a protein-coding gene, we mapped the consensus start determined from the alignment of homologous proteins (see above) to the query nucleotide sequence. When the consensus start was located upstream of the query start position, the nucleotide sequence was searched for in-frame start codons in between the original start position and the consensus starts. When several start codons were found in that region, the length of the extension of the query sequence was minimized, i.e., a new start position closest to the original annotated start was chosen. When the consensus start was located downstream of the query start position, the nucleotide sequence was searched for in-frame start codons located downstream of the annotated start. Again, when several start codons were found in that region, the loss of amino acids in the query sequence was minimized by selecting the start codon closest to the original start.

For each co-directional gene pair, the distance between the stop codon of the upstream gene and the start codon of the downstream gene was measured. Intergenic distance of 0 means that the genes are directly adjacent; a negative intergenic distance indicates an overlap. The genes were classified with respect to the distance from the upstream gene as: (1) "overlapping", with negative intergenic distance, (2) "close10", with the intergenic distance between 1 nt and 9 nt, (3) "medium50", with the intergenic distance is between 10 nt and 50 nt, (4) "far", with the intergenic distance greater than 50 nt and no known gene starting in the opposite strand at that location, and, (5) "lead", with intergenic distance greater than 50 nt and a known gene start on the opposite strand.

To identify the Shine-Dalgarno (SD) motif, we extracteded the 16S rRNA sequences of each species according to the GenBank annotations and searched their 3′ ends for the best match to the canonical SD motif, ACCTCC. All SD motifs were extended by 1 nt upstream and downstream to the length of 8 nt. To identify genes that employ the SD-dependent mechanism for translation initiation, the free2bind package[28] was used to determine the free energy ($\Delta G$) of base-pairing between the region 12 nt upstream from the translation start and the 3′ ends of the corresponding 16S RNAs. The genes with $\Delta G < -3.5$ kcal/mol[28] as using an SD signal to initiate translation, and genes with $\Delta G < -8.4$ kcal/mol were classified as using a strong SD signal[28].

**Reporting summary**. Further information on research design is available in the Nature Research Reporting Summary linked to this article.

## Data availability

The authors declare that all data supporting the findings of this study are available within the paper and its supplementary files. The average raw data to the quantification of specific enzyme activities and transcript levels are shown in Supplementary Figs. 3–6. The results for the single measurements are provided in the Source Data file. The results of the bioinformatics genome analyses are shown in Supplementary Data 1.

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

## Acknowledgements

We thank Svetlana Shabalina for expert help with the bioinformatics analysis. G.F., Y.I.W., and E.V.K. are supported by the Intramural Research Program of the National Institutes of Health of the USA, through the National Library of Medicine. The experimental part of the study was supported by the German Research Council (Deutsche Forschungsgemeinschaft) through grant No. So264/27-1.

## Author contributions

E.V.K. and J.S. conceived the study. G.F. and Y.I.W. performed the bioinformatics analyses. M.H., S.L., E.K., K.S., and C.W. performed the experiments. All authors analyzed the results. E.V.K. and J.S. wrote the paper.

## Additional information

**Competing interests:** The authors declare no competing interests.

