## [Peer Review File · Nature Communications]

Reviewers' comments:

Reviewer #1 (Remarks to the Author):

This paper describes a study at the genomic level on over 700 species of prokaryotes (archaea and bacteria), aimed at identifying the respective fractions of overlapping, co-transcribed genes. The working hypothesis was that such genes might be coupled translationally, i.e. translation of the upstream frame would be required to allow translation of the downstream one. Furthermore, the authors looked for the presence of Shine-Dalgarno motifs upstream of the overlapping genes, comparing these with "leading" genes, i.e. genes that come first in an operon, with the aim of understanding whether SD-motifs are required for efficient translational coupling.

The *in silico* predictions were then tested experimentally, studying the translational behavior of a number of chosen gene pairs of *Haloferax volcanii* as a representative of the Archaea and of *E.coli* as a representative of Bacteria. The relevant regulative regions of the chosen genes were inserted into reporter plasmids, and the translation products were identified and quantified.

The authors conclude that overlapping genes have a strong probability of being translationally coupled by a mechanism of termination/reinitiation. The importance of the SD-motifs for reinitiation at the downstream cistron appears to be highly variable and depending on the particular gene couple under analysis.

Upon the whole, the work is interesting, well done and contains valuable data that will be very useful to investigators working in the field, therefore it merits publication.

However, there are some concerns that the authors should consider.

-About the importance of SD-motifs for initiation/reinitiation, a certain caution should be applied in generalizing the results obtained with *H.volcanii* to the Archaea in general. As shown by the authors themselves, Haloarchaea deviate from the average pattern of internal-vs-leading SD motifs; moreover, they have a large proportion of leaderless mRNAs and make scarce, if any, use of SD-motifs for the translation of leading genes. I understand that performing *in vivo* translation experiments with other Archaea, which are much less tractable genetically than halophiles, would be rather difficult. However, there is at least another study in *Sulfolobus* that demonstrated independent translation of closely arranged genes in an operon, and an important role of internal SD motifs (Condò et al 1999 *Mol Microbiol.* 1999, 34(2):377-84). *Sulfolobus* belongs to the Crenarchaeota I (using the classification of this paper) which are very similar to Haloarchaea in having a scarce proportion of leading SD-motifs and a high proportion of leaderless mRNAs. Therefore I feel that this study should be quoted, if anything to emphasize that the usage of SD-motifs is indeed very idiosyncratic and that at the present stage one must be very cautious in making generalizations.

-Although the authors tend to suggest that gene overlapping means translational coupling, the study does not prove that this is true in all, or even in the generality, of cases (indeed, one of the chosen gene pairs in *E.coli* was not translationally coupled in this study). This must be kept in mind when discussing the putative evolutionary advantage of overlapping genes.

In the discussion, the authors put forward a number of hypotheses to explain why extensive translational coupling may be evolutionarily advantageous. I found them not entirely convincing.

For instance, why should two proteins be produced in exactly the same amount if there is termination/reinitiation? The speed/efficiency of elongation may be influenced by other factors, such as rare codons, sequence constraints (of both mRNA and protein) and so on.

In the same vein, the fact that operons may encode protein subunits that interact co-translationally does not strictly require translational coupling, since independently initiated proteins encoded by a same mRNA may well interact (and in fact they do).

Also, the sequence constraints given by internal SD also apply when there is no translational coupling but independent initiation, as it is often the case. Therefore, while having genes encoding related products in operons is surely advantageous, a full understanding of translational coupling will probably require further investigation. Maybe the authors should address these points in the

discussion.

Reviewer #2 (Remarks to the Author):

Huber et al. scrutinized 720 genomes of Archaea and Bacteria by means of bioinformatics for the presence of overlapping translation termination / re-initiation sites. It has been well established with several neighbouring gene pairs of Bacteria and accessory genetic elements that this genetic arrangement can lead to translational coupling of the downstream gene with the upstream gene. Moreover, it has been demonstrated that translational coupling can provide a means to ensure a certain stoichiometry of the corresponding gene products. The authors show for 14 archaeal and bacterial overlapping gene pairs that -with the exception of one genetic entity derived from *E. coli*- translation of the downstream gene depends on that of the upstream gene. In addition, they showed that replacement of the Shine-Dalgarno sequence of the downstream gene, which was present in all 14 gene overlaps, had variable effects in terms of translational output of the downstream genes. The molecular reasons for this variability have not been further assessed. Although the claims of the paper are not completely novel, a similar comprehensive study has so far to my knowledge not been performed. In particular, there are no reports on translational coupling in Archaea. The work of the authors is mostly technically sound but needs corrections and a re-interpretation at some places (see below). The results are generally discussed in the context of the relevant literature. Nevertheless, the authors should include and discuss their results also in the context of Levin-Karo et al. (2013), who have used fluorescent reporters to study translational coupling in *E. coli*. Moreover, as specified below some parts could be more succinctly written and the "Discussion" could be shortened by omitting reiterations from the previous sections.

1. L. 36-38: This sentence appears to be too generic as a detailed discussion on different "strategies" is missing.
2. Figure 1: The drawing could be provided in a more space saving and elegant fashion! In addition, the text in the legend should be more self-explaining.
3. Page 3, line 69: The *drxAB* operon is derived from *Streptomyces peucetius* and not from *E. coli*!
4. Page 4, lines 84-91; Fig. 1C: Fig. 1C does not really show the scenario that is described in the text!
5. L. 94-91: Although analogous to Bacteria, re-initiation events in eukaryotic RNA viruses appear to be mechanistically distinct from 70S re-initiation events as proposed by the Nierhaus group. Therefore, I would suggest to delete lines 84-91 and continue with line 92.
6. L.133: The authors seem to mix up classes and orders. Please use appropriate systematic nomenclature.
7. Fig. 2B could be provided as a supplementary Fig.
8. L.157/158: pls. omit first part of the sentence as independent translation initiation without a SD is counterintuitive.
9. Fig. 3, legend: Needs a better explanation why the SD is "more important" for either genetic entity!
10. Fig. 4 could be provided in the supplements as no further experiments derive from these analyses.
11. Figs. 5 and 6: The genes should be termed correctly and given in italics. For h.p., the orf should be provided.
12. Fig. 5C/D experiments: Why did the authors use different gene pairs when compared to Fig 5A/B? To assess any influence of the 5' coding region on translational coupling, the authors need to provide the corresponding constructs for the pairs they have studied in Fig. 5A/B. The experiments with the constructs shown in Fig 5C/D could be used as further evidence and provided in the supplements. I'm also wondering about the negative "normalized values" (Fig. 5B/D). The normalized translational efficiency can at best be zero but not negative, no matter what the software is telling!
13. Fig. 5F: The interpretation observed with the *hyfHI* entity is not straightforward. Obviously the activity or production of *GlpD* is affected by part X.

14. Fig. 5G; bioBF: The bioF gene does not seem to have a canonical SD sequence. Could this be the reason why no effect is observed on gusA translation after introduction of the stop codon? The statement in lines 279-280 is pure speculation. The authors may either elaborate on this or this result should be omitted.

15. Fig. 6C/D: Again, I'm wondering why the authors have not used the same genetic set ups as shown in Fig. 6A!

16. Fig. 6G: The bioBF entity does not behave as anticipated. Again, there appears to be a lack of a canonical SD motif for bioF!

17. Discussion: As mentioned above, the authors are requested to streamline this section and to remove redundancies. E.g: Page 12: Statements on the SD motif; Page 13, lines 395-400;

18. L411-422: Masking of the downstream SD sequences by structure may be too simplistic. This would also depend on the translation frequency of the upstream gene when compared to the downstream gene. In other words, if fast translation of the upstream gene occurs, there might be no independent binding of ribosomes to the downstream SD.

Reviewer #3 (Remarks to the Author):

In this study, the authors examined the phenomenon of co-directed overlapping genes. They show that such gene pairs are prevalent in bacterial and archaeal genomes, and test the hypothesis that these pairs undergo coupled translation. For 13 out of 14 tested pairs, the authors provide evidence for translational coupling that depends on start and stop codon overlap. They further examine the contribution of Shine-Dalgarno motifs in the upstream genes on the translation efficiency of the downstream genes.

Overlapping genes have been previously described, but here the authors provide new and interesting insights regarding the role of overlapping codons and SD motifs on coupled translation of gene pairs in their native context while talking into account both RNA and protein levels. I expect these results to be of interest to the general scientific community. Below are minor issues that should be addressed prior to publication:

- Consider citing in the Introduction the work by Johnson and Chisholm from 2004, which describe a large-scale computational analysis of overlapping genes in microbial genomes.
- The exact nature of the overlapping pair examined across all genomes should be better defined in the first paragraph of the results section (currently: "co-directional overlapping gene pairs with potential for translational coupling").
- Please provide in the methods section the exact criteria applied to select the 720 representative bacterial genomes. Also, please include in Supp. Table S1, the accession numbers of the genomes to facilitate reproducibility.
- The experiments to determine the role of SD motifs in *H. volcanii* were performed in seven of the nine gene pairs that were previously tested for coupled translation. Please specify why these seven were chosen and the other two gene pairs were omitted.
- It would be beneficial if the authors could speculate why for the rpl30/rpl15 pair mutation in the SD motif resulted with a considerable increase in translation efficiency of the downstream gene.
- When reporting that removal of SD motif for the bioBF pair had no effect on the translation efficiency of the downstream gene (line 320), it is worth mentioning that this gene pair does not seem to undergo translation coupling (as reported in the previous sub-section).
- Line 376: It is not clear which 3 archaeal groups and 3 bacterial groups the sentence refers to.

According to Figure 4C, all six archaeal groups have a SD re-initiation to de novo initiation quotient > 2 . There are indeed only three archaeal groups with quotient >4 , but none of the bacterial groups presented are above this threshold. Please settle this by elaborating on the cutoff used and by specifying the relevant archaeal and bacterial groups.

- Lines 413–420 in the Discussion contain the first mention of results regarding mRNA folding analysis that. Consider moving these to the Results section.
- I find the Supplementary Methods include information important to the understanding of the research (including the identification of SD signals, translation start site and definition of “leading” genes. Unless there a acute space consideration, please include the Supplementary methods as part of the main text Method section.
- The text describing the process of protein homolog identification in the supplementary method (lines 12-34) would greatly benefit from a diagram presenting the process visually.
- In the “Identification of protein homologs” the motivation for some of the steps is not clear, for example in lines 24-26: “For each LLC, we compared proteins from rep720 against the loose cluster blast databases they belong to, using blastp software and selecting hits with an e-value $\leq 10^{-4}$ ”. The purpose of this stage is unclear, and it is not specified to what exactly these hits were selected.
- Some of the figure legends would benefit from additional information. For example: in Figure 2A specify the type of overlapping genes presented (unless any overlap is considered), in Figure 3A and 4A note that the fraction is out of overlapping genes, and in Figure 5 mention how the values were normalized.

Phrasing/technical:

- In the abstract, the authors mention they identified a “substantial, albeit highly variable fraction of co-directed overlapping genes”. Since this is such a major part of this study, I suggest mentioning already in the Abstract the percentage of overlapping genes in prokaryotes (as described in the first paragraph of the Results)
- Line 169: unnecessary hyphen.
- Lines 181–2: Please rephrase “the fractions with strong SD motifs were considerably lower than those with SD” (both are with SD motifs, the contrast is between “normal” SD motifs and strong ones.
- Line 324: please specify that the statement refers specifically to *E. coli*.
- Line 344: double stop at the end of the sentence.
- Line 371: The sentence “The remaining groups of prokaryotes deviate more or less from the four patterns described above” is unclear, please rephrases.
- I suggest slightly dialing down a bit the sentence in line 401: “... we characterized native gene pairs of *E.coli*” to reflect the fact the experiments were not performed on complete native genes but rather on constructs containing native gene fragments.
- Line 445: consider changing: “...70S ribosome can stay intact, can linear scan the mRNA and reinitiate...” to “...70S ribosome can stay intact, linearly scan the mRNA, and reinitiate...”

- In figures 2–4 commas are used as decimal separators, while in the text percentage are used, and in figures 5–6 a decimal dot is used. I suggest being consistent across the manuscript (I think percentages would be preferable figures 2–4).
- Supp. methods, line 38: “starts position” should be “start positions”.

Response to the reviewers comments to the manuscript entitled “Translational coupling via termination-reinitiation in archaea and bacteria”

Reviewers' comments (in black) and responses (in red)

Reviewer #1 (Remarks to the Author):

This paper describes a study at the genomic level on over 700 species of prokaryotes (archaea and bacteria), aimed at identifying the respective fractions of overlapping, co-transcribed genes. The working hypothesis was that such genes might be coupled translationally, i.e. translation of the upstream frame would be required to allow translation of the downstream one. Furthermore, the authors looked for the presence of Shine-Dalgarno motifs upstream of the overlapping genes, comparing these with “leading” genes, i.e. genes that come first in an operon, with the aim of understanding whether SD-motifs are required for efficient translational coupling.

The in silico predictions were then tested experimentally, studying the translational behavior of a number of chosen gene pairs of *Haloferax volcanii* as a representative of the Archaea and of *E.coli* as a representative of Bacteria. The relevant regulative regions of the chosen genes were inserted into reporter plasmids, and the translation products were identified and quantified.

The authors conclude that overlapping genes have a strong probability of being translationally coupled by a mechanism of termination/reinitiation. The importance of the SD-motifs for reinitiation at the downstream cistron appears to be highly variable and depending on the particular gene couple under analysis.

Upon the whole, the work is interesting, well done and contains valuable data that will be very useful to investigators working in the field, therefore it merits publication.

Thank you for this positive opinion.

However, there are some concerns that the authors should consider.

-About the importance of SD-motifs for initiation/reinitiation, a certain caution should be applied in generalizing the results obtained with *H.volcanii* to the Archaea in general. As shown by the authors themselves, Haloarchaea deviate from the average pattern of internal-

vs-leading SD motifs; moreover, they have a large proportion of leaderless mRNAs and make scarce, if any, use of SD-motifs for the translation of leading genes. I understand that performing in vivo translation experiments with other Archaea, which are much less tractable genetically than halophiles, would be rather difficult. However, there is at least another study in *Sulfolobus* that demonstrated independent translation of closely arranged genes in an operon, and an important role of internal SD motifs (Condò et al 1999 Mol Microbiol. 1999, 34(2):377-84). *Sulfolobus* belongs to the Crenarchaeota I (using the classification of this paper) which are very similar to Haloarchaea in having a scarce proportion of leading SD-motifs and a high proportion of leaderless mRNAs. Therefore I feel that this study should be quoted, if anything to emphasize that the usage of SD-motifs is indeed very idiosyncratic and that at the present stage one must be very cautious in making generalizations.

Thank you for the reference and the advice not to over-generalize. The *Sulfolobus* study has been integrated into the manuscript. At several places sentences have been rephrased to be more cautious about generalization, and the word “universal” is no longer used in the context of translational coupling.

-Although the authors tend to suggest that gene overlapping means translational coupling, the study does not prove that this is true in all, or even in the generality, of cases (indeed, one of the chosen gene pairs in *E.coli* was not translationally coupled in this study). This must be kept in mind when discussing the putative evolutionary advantage of overlapping genes. In the discussion, the authors put forward a number of hypotheses to explain why extensive translational coupling may be evolutionarily advantageous. I found them not entirely convincing. For instance, why should two proteins be produced in exactly the same amount if there is termination/reinitiation? The speed/efficiency of elongation may be influenced by other factors, such as rare codons, sequence constraints (of both mRNA and protein) and so on.

It is true that the fraction of overlapping genes that are translationally coupled is unknown. The number of studied cases is extremely low, translational coupling has mostly been studied using non-overlapping genes on bicistronic transcripts. However, until now I know only two counter-examples, i.e. the *bioBF* gene couple described in this manuscript and the *Sulfolobus* genes mentioned above (which are not really overlapping, but have adjacent stop-start codons). We have added to the discussion that the mentioned possible advantages could only

apply when the efficiency of translational coupling is about 100%, and that the efficiency of coupling is unknown. In my opinion the elongation speed does not influence the amount of produced protein, because initiation is the rate-limiting step (unless elongation is so slow at the 5'-end that it decreases the initiation frequency).

In the same vein, the fact that operons may encode protein subunits that interact co-translationally does not strictly require translational coupling, since independently initiated proteins encoded by a same mRNA may well interact (and in fact they do).

We fully agree that many examples of non-coupled polycistronic mRNAs exist that encode subunits of heteromeric complexes. This point is addressed in the Introduction as well as in the Discussion.

Also, the sequence constraints given by internal SD also apply when there is no translational coupling but independent initiation, as it is often the case. Therefore, while having genes encoding related products in operons is surely advantageous, a full understanding of translational coupling will probably require further investigation. Maybe the authors should address these points in the discussion.

It is true that sequence constraints also apply for internal SDs and non-coupled gene pairs, however, we think that these cases are rather seldom. In our opinion in many cases the intergenic distance between genes on polycistronic mRNAs is larger than 10 nt, and thus the SD does not put any constraint on the protein sequence of the upstream gene. We have added that further investigation is needed for a full understanding of translational coupling.

Reviewer #2 (Remarks to the Author):

Huber et al. scrutinized 720 genomes of Archaea and Bacteria by means of bioinformatics for the presence of overlapping translation termination / re-initiation sites. It has been well established with several neighbouring gene pairs of Bacteria and accessory genetic elements that this genetic arrangement can lead to translational coupling of the downstream gene with the upstream gene. Moreover, it has been demonstrated that translational coupling can provide a means to ensure a certain stoichiometry of the corresponding gene products. The authors show for 14 archaeal and bacterial overlapping gene pairs that -with the exception of one genetic entity derived from E. coli- translation of the downstream gene depends on that of the

upstream gene. In addition, they showed that replacement of the Shine-Dalgarno sequence of the downstream gene, which was present in all 14 gene overlaps, had variable effects in terms of translational output of the downstream genes. The molecular reasons for this variability have not been further assessed. Although the claims of the paper are not completely novel, a similar comprehensive study has so far to my knowledge not been performed. In particular, there are no reports on translational coupling in Archaea. The work of the authors is mostly technically sound but needs corrections and a re-interpretation at some places (see below). The results are generally discussed in the context of the relevant literature. Nevertheless, the authors should include and discuss their results also in the context of Levin-Karo et al. (2013), who have used fluorescent reporters to study translational coupling in *E. coli*. Moreover, as specified below some parts could be more succinctly written and the “Discussion” could be shortened by omitting reiterations from the previous sections.

Thank you for the positive opinion about our study.

Thank you for pointing out the elegant study by Levin-Karp that we had overlooked, it has now been integrated. The discussion has been shortened.

1. L. 36-38: This sentence appears to be too generic as a detailed discussion on different “strategies” is missing.

The sentence has been changed. “Discussed strategies” has been replaced by “observed patters”, which better fits to our discussion.

2. Figure 1: The drawing could be provided in a more space saving and elegant fashion! In addition, the text in the legend should be more self-explaining.

The size has been reduced, so that the Figure now fits a single column. A third panel has been added to the Termination-Reinitiation part (1C) to better visualize that the same 30S (or 70S) translates both genes. The text of the legend has been changed to be more self-explaining.

3. Page 3, line 69: The *drrAB* operon is derived from *Streptomyces peucetius* and not from *E. coli*!

Sorry for this mistake. The *drrAB* operon has been removed.

4. Page 4, lines 84-91; Fig. 1C: Fig. 1C does not really show the scenario that is described in the text!

A further panel has been added to Fig. 1C to visualize that the same 30S (or 70S) ribosome, which had translated the upstream gene, also translates the downstream gene. Also, a yellow bar representing the SD motif has been added. In addition, the reference to Fig. 1C has been moved to a more appropriate position.

5. L. 94-91: Although analogous to Bacteria, re-initiation events in eukaryotic RNA viruses appear to be mechanistically distinct from 70S re-initiation events as proposed by the Nierhaus group. Therefore, I would suggest to delete lines 84-91 and continue with line 92. In our opinion the Nierhaus model of 70S scanning is highly controversial, and experimental evidence contradicting 70S scanning also exists. Therefore, we would like to leave the discussion of termination-reinitiation via 30S subunit versus via 70S ribosomes open, and we would like to introduce both possibilities.

6. L.133: The authors seem to mix up classes and orders. Please use appropriate systematic nomenclature.

It is true that the “groups” analyzed and discussed in the bioinformatics analysis (Figures 2, 3, and S2) are not on the same systematic level. However, if the same systematic level would be used, important findings would be missed, e.g. the very different patterns of two groups of Crenarchaeota, the large difference between halophilic archaea and methanogenic archaea, the variations within the proteobacteria, etc. Therefore, we kept the “groups” that are not all on an identical systematic level.

7. Fig. 2B could be provided as a supplementary Fig.

Fig. 2b has been moved to the Supplement, as proposed (new Fig. S1).

8. L.157/158: pls. omit first part of the sentence as independent translation initiation without a SD is counterintuitive.

Independent translation initiation without a SD might indeed be counterintuitive and against the textbooks, but, in fact, it massively occurs. Even in *E. coli* only 60-70% of all 5'-UTRs contain a SD motif, and thus 30-40% have independent translation initiation in the absence of a SD. In Bacteroidetes SD motifs do not operate in 5'-UTRs and thus initiation at all transcripts is SD-independent. If translation initiation at 5'-UTRs is SD-independent, in our opinion it cannot be ruled out that this could also happen at downstream genes (of course, in both cases the molecular mechanism is unclear). Therefore, we kept the sentence.

9. Fig. 3, legend: Needs a better explanation why the SD is “more important” for either genetic entity!

“More important” has been replaced with “more frequently used”.

10. Fig. 4 could be provided in the supplements as no further experiments derive from these analyses.

Fig. 4 has been moved to the supplementary material, as proposed (new Fig. S2).

11. Figs. 5 and 6: The genes should be termed correctly and given in italics. For h.p., the orf should be provided.

Good suggestion, has been done.

12. Fig. 5C/D experiments: Why did the authors use different gene pairs when compared to Fig 5A/B? To assess any influence of the 5' coding region on translational coupling, the authors need to provide the corresponding constructs for the pairs they have studied in Fig. 5A/B. The experiments with the constructs shown in Fig 5C/D could be used as further evidence and provided in the supplements.

Four additional genes were used to increase the number of analyzed genes from 5 to 9. The results presented in Fig. 5D (old) clearly show that translational coupling occurs in the absence of native downstream sequences, in all four cases. The *dhfr* reporter gene is monocistronic and, thus, it can be excluded that it carries downstream gene motifs. We think that it is important to show all 9 examples in the main text, and, therefore, we have not moved the first five or the second four examples to the Supplement.

I'm also wondering about the negative “normalized values” (Fig. 5B/D). The normalized translational efficiency can at best be zero but not negative, no matter what the soft-ware is telling!

We fully agree that the real translation efficiency cannot be negative, this does not make physical sense. However, as good biochemists we performed a negative control (plasmid lacking gene x and the reporter gene), and, as usual, subtracted the negative control from all test values. In cases when the test values are zero, subtraction of two values that are based on physical identical enzyme levels (zero) results in values that are small and can be either

positive or negative. We think that all values should be calculated in the same way, which is standard in biochemical experiments.

13. Fig. 5F: The interpretation observed with the *hyfHI* entity is not straightforward.

Obviously the activity or production of *GlpD* is affected by part X.

We share the interpretation of the reviewer that in this case *GlpD* is affected by part X. We have now added this to the text. Translational fusions to reporter genes always have the risk that the fusion protein might not have the identical specific activity than the reporter protein alone. However, in many cases reporter genes are very informative, and, thus, they are very widely used.

14. Fig. 5G; *bioBF*: The *bioF* gene does not seem to have a canonical SD sequence. Could this be the reason why no effect is observed on *gusA* translation after introduction of the stop codon? The statement in lines 279-280 is pure speculation. The authors may either elaborate on this or this result should be omitted.

We have deliberately chosen *E. coli* genes with SD motifs of different strengths. In our opinion the absence (!) of a good SD motif cannot explain the high (!) translational efficiency in the stop codon variant. We agree that lines 279-280 is speculative, and we have now replaced “It seems possible” with “One theoretically possible explanation”.

15. Fig. 6C/D: Again, I’m wondering why the authors have not used the same genetic set ups as shown in Fig. 6A!

If we have used the same genes X in Fig. 6C/D than in 6A/B, we would have mutated the identical SD motifs twice. Instead, we have mutated SD motifs in different genes X in 6D versus 6B, thereby increasing the data base.

16. Fig. 6G: The *bioBF* entity does not behave as anticipated. Again, there appears to be a lack of a canonical SD motif for *bioF*!

Replacement of the very weak SD was not really necessary, because the total removal of coupling resulted in a high translational efficiency (old Fig. 5G, now 4G). Therefore, in our opinion the result for *bioFB* shown in 6G (now 5G) could be expected after the result shown in 5G (now 4G), but it is not informative, in contrast to the results of the other four genes shown in 6G.

17. Discussion: As mentioned above, the authors are requested to streamline this section and to remove redundancies. E.g: Page 12: Statements on the SD motif; Page 13, lines 395-400; **The statements to the SD motif on page 12 as well as lines 395 – 400 have been considerably shortened.**

18. L411-422: Masking of the downstream SD sequences by structure may be too simplistic. This would also depend on the translation frequency of the upstream gene when compared to the downstream gene. In other words, if fast translation of the upstream gene occurs, there might be no independent binding of ribosomes to the downstream SD.

In our opinion masking of the SD motif by structure has very different outcomes for long-rang interactions (new ribosomes bind, higher efficiency of downstream gene possible) in contrast to local structures around the start codon (termination-reinitiation, the same ribosome for both genes). Because SD motifs attract 30S subunits in 5'-UTRs and at operon genes with intergenic regions in *E. coli*, there should be an explanation why they fail to do so at overlapping gene pairs. To us, local structures seem to be the most plausible possibility.

Reviewer #3 (Remarks to the Author):

In this study, the authors examined the phenomenon of co-directed overlapping genes. They show that such gene pairs are prevalent in bacterial and archaeal genomes, and test the hypothesis that these pairs undergo coupled translation. For 13 out of 14 tested pairs, the authors provide evidence for translational coupling that depends on start and stop codon overlap. They further examine the contribution of Shine-Dalgarno motifs in the upstream genes on the translation efficiency of the downstream genes.

Overlapping genes have been previously described, but here the authors provide new and interesting insights regarding the role of overlapping codons and SD motifs on coupled translation of gene pairs in their native context while talking into account both RNA and protein levels. I expect these results to be of interest to the general scientific community. **Thank you for the opinion that our results will be of general interest.**

Below are minor issues that should be addressed prior to publication:

- Consider citing in the Introduction the work by Johnson and Chisholm from 2004, which describe a large-scale computational analysis of overlapping genes in microbial genomes.

Thank you for the reference, it has been integrated.

- The exact nature of the overlapping pair examined across all genomes should be better defined in the first paragraph of the results section (currently: “co-directional overlapping gene pairs with potential for translational coupling”).

All lengths of overlaps were included, this is explicitly indicated in the revision.

- Please provide in the methods section the exact criteria applied to select the 720 representative bacterial genomes. Also, please include in Supp. Table S1, the accession numbers of the genomes to facilitate reproducibility.

In the revised manuscript, the selection criteria are specified, and the accession numbers are included in table S1.

- The experiments to determine the role of SD motifs in *H. volcanii* were performed in seven of the nine gene pairs that were previously tested for coupled translation. Please specify why these seven were chosen and the other two gene pairs were omitted.

The reason is very trivial: a student thesis that concentrated on the role of the SD motif in *H. volcanii* ended (the time limit was reached) as the first seven mutants had been generated and characterized, and the last two examples had not yet been started. Already the first seven examples clearly showed that the effect of the SD motif is gene-specific and very variable, therefore, a very time-consuming new project for the last two examples was not started.

- It would be beneficial if the authors could speculate why for the rpl30/rpl15 pair mutation in the SD motif resulted with a considerable increase in translation efficiency of the downstream gene.

This result was very surprising to us and we do not have an explanation. If the SD motif would have been very long with 8-10 nt, a possible explanation could have been that clearance of the start codon takes so long that the reinitiation frequency is low in the wildtype. However, the SD motif has an ideal length of 6 nt. Therefore, unfortunately, we do not have any plausible putative explanation that we could add as a speculation.

- When reporting that removal of SD motif for the bioBF pair had no effect on the translation efficiency of the downstream gene (line 320), it is worth mentioning that this gene pair does not seem to undergo translation coupling (as reported in the previous sub-section).

Good point, this information has been added.

- Line 376: It is not clear which 3 archaeal groups and 3 bacterial groups the sentence refers to. According to Figure 4C, all six archaeal groups have a SD re-initiation to de novo initiation quotient > 2 . There are indeed only three archaeal groups with quotient >4 , but none of the bacterial groups presented are above this threshold. Please settle this by elaborating on the cutoff used and by specifying the relevant archaeal and bacterial groups.

The aim was to refer to the SD motif and not to the strong SD motif, therefore, the reference to Figure 4C was misleading, it should have been Figure 3C. A cutoff of 1.5 was applied to reach the conclusion, and this is now explained in the text.

- Lines 413–420 in the Discussion contain the first mention of results regarding mRNA folding analysis that. Consider moving these to the Results section.

The in silico folding analyses of putative local RNA structures have been moved to the Results section, as proposed.

- I find the Supplementary Methods include information important to the understanding of the research (including the identification of SD signals, translation start site and definition of “leading“ genes. Unless there a acute space consideration, please include the Supplementary methods as part of the main text Method section.

Given that there are no space restrictions for Methods, we followed this recommendation and moved all computational methods into the main body of the manuscript, with appropriate editing to eliminate redundancy.

- The text describing the process of protein homolog identification in the supplementary method (lines 12-34) would greatly benefit from a diagram presenting the process visually.

A flowchart of the process is included in the revision (Supplementary figure S8).

- In the “Identification of protein homologs” the motivation for some of the steps is not clear, for example in lines 24-26: “For each LLC, we compared proteins from rep720 against the

loose cluster blast databases they belong to, using blastp software and selecting hits with an e-value $\leq 10^{-4}$ ". The purpose of this stage is unclear, and it is not specified to what exactly these hits were selected.

We clarified the method sections and include the detailed pipeline as Sup Figure S8. Briefly, for each LLC, we created a blast database, extracted the protein from rep720 to form queries, and ran blastp with these queries against the LLC blast database. The main purpose is to select a set of 10 most confident homologs – the best 10 significant hits according to the blast e-value.

- Some of the figure legends would benefit from additional information. For example: in Figure 2A specify the type of overlapping genes presented (unless any overlap is considered), in Figure 3A and 4A note that the fraction is out of overlapping genes, and in Figure 5 mention how the values were normalized.

Indeed, any overlap is considered as explicitly stated in the revised Results. The additional information has been added to the Figure legends as suggested.

Phrasing/technical:

- In the abstract, the authors mention they identified a “substantial, albeit highly variable fraction of co-directed overlapping genes”. Since this is such a major part of this study, I suggest mentioning already in the Abstract the percentage of overlapping genes in prokaryotes (as described in the first paragraph of the Results)

The average fraction of overlapping genes is included in the revised Abstract as suggested.

- Line 169: unnecessary hyphen.

Removed

- Lines 181–2: Please rephrase “the fractions with strong SD motifs were considerably lower than those with SD” (both are with SD motifs, the contrast is between “normal” SD motifs and strong ones).

Rephrased for clarity: “For overlapping gene pairs, the strong SD motifs represented a relatively small fraction of all identified instances of SD...”

- Line 324: please specify that the statement refers specifically to E. coli.

Has been added.

- Line 344: double stop at the end of the sentence.

Has been removed.

- Line 371: The sentence “The remaining groups of prokaryotes deviate more or less from the four patterns described above” is unclear, please rephrases.

The sentence has been removed.

- I suggest slightly dialing down a bit the sentence in line 401: “... we characterized native gene pairs of E.coli” to reflect the fact the experiments were not performed on complete native genes but rather on constructs containing native gene fragments.

The information has been added that not whole genes, but native gene fragments were used.

- Line 445: consider changing: “...70S ribosome can stay intact, can linear scan the mRNA and reinitiate...” to “...70S ribosome can stay intact, linearly scan the mRNA, and reinitiate...”

The whole paragraph has been removed.

- In figures 2–4 commas are used as decimal separators, while in the text percentage are used, and in figures 5–6 a decimal dot is used. I suggest being consistent across the manuscript (I think percentages would be preferable figures 2–4).

The fraction values in Figures 2-4 have been transformed into % values, to be consistent between text and Figures and to follow the valuable suggestion.

- Supp. methods, line 38: “starts position” should be “start positions”.

Corrected

REVIEWERS' COMMENTS:

Reviewer #1 (Remarks to the Author):

I find the revised manuscript substantially improved as the authors were able to answer satisfactorily most of the points raised. Only one thing regarding the role of SD motifs: as they state in the answer letter, the authors did discuss the previous study on *Sulfolobus* (lines 362-371) but the quotation is missing. Please insert it.

Reviewer #3 (Remarks to the Author):

The authors have fully addressed all of my concerns.

I have a couple of minor technical comments:

1) Parts of the new Supplemental Figure S8 appears in low resolution (at least in the merged PDF). I hope this is just a technical problem with the conversion. Please make sure the final figure is in high resolution and all sections readable.

2) Line 557: "and the normalized relative transcript levels." should be rephrased. Probably should be "relative **to** transcript levels" and either "and **then** normalized" or "and the **values** normalized"

REVIEWERS' COMMENTS: and our answers to the comments

Reviewer #1 (Remarks to the Author):

I find the revised manuscript substantially improved as the authors were able to answer satisfactorily most of the points raised. Only one thing regarding the role of SD motifs: as they state in the answer letter, the authors did discuss the previous study on *Sulfolobus* (lines 362-371) but the quotation is missing. Please insert it.

I apologize that I forgot to insert the reference. Thank you for noticing the missing reference. It has been included.

Reviewer #3 (Remarks to the Author):

The authors have fully addressed all of my concerns.

I have a couple of minor technical comments:

1) Parts of the new Supplemental Figure S8 appears in low resolution (at least in the merged PDF). I hope this is just a technical problem with the conversion. Please make sure the final figure is in high resolution and all sections readable.

It was indeed a problem of the conversion between different formats. A high resolution version of Supplementary Figure 8 has been included in the final submission.

2) Line 557: "and the normalized relative transcript levels." should be rephrased. Probably should be "relative *to* transcript levels" and either "and *then* normalized" or "and the *values* normalized"

The sentence has been rephrased.